# SPECTRA: Spectral Target-Aware Graph Augmentation for Imbalanced Molecular Property Regression

## Abstract

In molecular property prediction, the most valuable compounds (e.g., high potency) often occupy sparse regions of the target space. Standard Graph Neural Networks (GNNs) commonly optimize for the average error, underperforming on these uncommon but critical cases, with existing oversampling methods often distorting molecular topology. In this paper, we introduce SPECTRA, a Spectral Target-Aware graph augmentation framework that generates realistic molecular graphs in the spectral domain. SPECTRA (i) reconstructs multi-attribute molecular graphs from SMILES; (ii) aligns molecule pairs via (Fused) Gromov–Wasserstein couplings to obtain node correspondences; (iii) interpolates Laplacian eigenvalues/eigenvectors and node features in a stable shared basis; and (iv) reconstructs edges to synthesize physically plausible intermediates with interpolated targets. A rarity-aware budgeting scheme, derived from a kernel density estimation of labels, concentrates augmentation where data are scarce. Coupled with a spectral GNN using edge-aware Chebyshev convolutions, SPECTRA densifies underrepresented regions without degrading global accuracy. On benchmarks, SPECTRA consistently improves error in relevant target ranges while maintaining competitive overall MAE, and yields interpretable synthetic molecules whose structure reflects the underlying spectral geometry. Our results demonstrate that spectral, geometry-aware augmentation is an effective and efficient strategy for imbalanced molecular property regression.

## 1 Introduction

Graph-structured data plays a central role in many scientific domains, including drug discovery, materials science, and genomics. These fields produce large volumes of complex, structured information that can be naturally represented as graphs, where nodes correspond to entities (e.g., atoms, molecules, genes) and edges capture their relationships (e.g., chemical bonds, interactions). Graph Neural Networks (GNNs) have transformed the modeling of such data by operating directly on graph structures, enabling state-of-the-art predictions of molecular properties, material characteristics, and biological interactions. In drug discovery, for example, GNNs have been applied to property prediction (Xiong et al., 2020), molecular design (Jin et al., 2018), and drug–target interaction prediction (Lim et al., 2019), with increasing adoption by the pharmaceutical industry to accelerate a development pipeline that typically exceeds $1 billion and a decade of effort (Vamathevan et al., 2019). Similar advances have been reported in materials science, where GNNs help identify compounds with desirable structural and functional properties (Karamad et al., 2020).

Despite this progress, a fundamental challenge remains largely unsolved: *imbalanced regression* on graphs. While imbalanced classification has received significant attention in graph learning, the regression setting has been comparatively neglected (Almeida et al., 2024; Xia et al., 2024; Ribeiro & Moniz, 2020a; Liu et al., 2023b). Yet, many scientific problems involve continuous targets where the most valuable outcomes are rare. Standard GNNs and other machine learning methods typically optimize for average performance across the full label distribution, which leads to poor accuracy in these rare but scientifically important regions. Classical oversampling techniques can mitigate imbalance but often fail to preserve the intricate topological and chemical properties of molecular graphs, limiting their practical effectiveness.

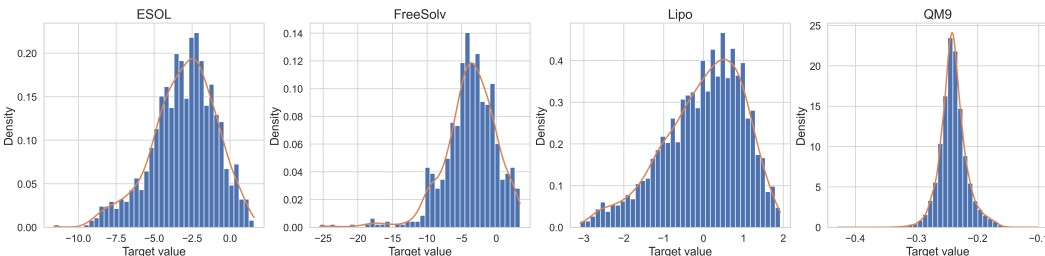

Figure 1: Distribution of target property values across three molecular datasets (ESOL, FreeSolv, Lipo and QM9 (homo property). Each subplot shows a normalized histogram of the experimental values with a Gaussian kernel density estimate (KDE) overlaid using Scott's rule-of-thumb bandwidth. These plots highlight the skewness and spread of target distributions, which can influence model training and performance.

Another limitation lies in *embedding-based augmentation methods*, which often generate synthetic molecules in latent spaces that lack interpretability and offer no guarantees of structural validity. As a result, the augmented samples may not correspond to chemically realistic molecules, hindering trust and practical adoption.

To address these challenges, we propose **SPECTRA** – *Spectral Target-Aware Graph Augmentation for Imbalanced Molecular Property Regression*. SPECTRA introduces a novel approach to oversampling that operates directly in the spectral domain of graphs. Specifically, it leverages the eigenspace of the graph Laplacian to interpolate both Laplacian spectra and node features of matched graphs in a shared spectral basis. This process produces synthetic molecular graphs that are structurally coherent, chemically plausible, and explicitly tailored to underrepresented regions of the target distribution 1. Unlike black-box embedding methods, SPECTRA provides interpretability by generating realistic molecules whose structures can be directly examined, while achieving significantly lower computational cost compared to existing state-of-the-art techniques.

Our contributions can be summarized as follows:

- **Novel methodology.** We introduce a spectral augmentation framework that augments samples in low-density regions of the label space while preserving topological fidelity, overcoming the limitations of existing oversampling techniques in regression.
- **Improved predictive performance.** Across benchmark molecular property datasets, SPECTRA achieves low error on rare compounds without degrading performance on common cases.
- **Interpretability and efficiency.** The synthetic graphs generated by SPECTRA are realistic and chemically meaningful, enabling direct inspection of augmented molecules while maintaining a lower computational footprint compared to competing approaches.

Together, these findings demonstrate that spectral graph augmentation is an effective and interpretable strategy for tackling imbalanced regression in molecular property prediction. The code and dataset are available in `https://anonymous.4open.science/r/SPECTRA-0D3C`

## 2 RELATED WORK

The challenge of imbalanced distributions in graph learning tasks has received increasing attention, particularly in scientific domains where rare values are critical. Recent research by Almeida et al. (2024) demonstrates that imbalanced learning in drug discovery datasets can be tackled with techniques such as oversampling and loss function manipulation when using Graph Neural Networks (GNNs). Despite these advances, most approaches operate directly in graph space rather than the spectral domain, limiting their ability to maintain global structural constraints. Bo et al. (2023b) published a comprehensive survey on spectral GNNs, highlighting their unique ability to capture global information and provide better expressiveness than spatial approaches. Wang & Zhang (2022) further analyzed the theoretical expressive power of spectral GNNs, proving that they can produce

arbitrary graph signals under specific conditions. However, these methods focus on balanced and classification datasets, illustrating the novelty and significance of SPECTRA.

## 2.1 Imbalanced Learning

Class imbalance has traditionally been addressed through resampling strategies, such as under-sampling majority classes or over-sampling minority classes. SMOTE (Chawla et al., 2002), for instance, generates synthetic minority samples by interpolating labeled data. Alternative approaches include cost-sensitive learning (Cui et al., 2019; Lin et al., 2017), which increases the loss weight of minority classes, and posterior re-calibration (Cao et al., 2019; Menon et al., 2020; Tian et al., 2020), which encourages larger margins for minority predictions.

Imbalanced regression introduces additional challenges because the labels are continuous rather than categorical (Ribeiro & Moniz, 2020a). Several methods from classification have been adapted to this setting. For example, SMOGN Branco et al. (2017) extends SMOTE to regression, while BMSE Ren et al. (2022) adapts logit re-calibration for numerical targets. LDS Yang et al. (2021) smooths the label distribution using kernel density estimation, and RankSim Gong et al. (2022) regularizes the latent space by aligning distances in label and feature space. Other approaches include SERA Ribeiro & Moniz (2020b), which proposes a relevance-aware evaluation metric; SGIR Liu et al. (2023a), which leverages unlabeled graphs to enrich underrepresented label ranges; and SIRN Zong et al. (2024), which combines deviation modeling with adaptive pseudo-label selection. While these methods improve performance in underrepresented regions, they often reduce accuracy in well-represented areas, especially under limited supervision or when relying heavily on pseudo-labeling.

## 2.2 Spectral Graph Methods

Spectral graph theory has applications spanning dimensionality reduction, clustering, and graph signal processing. Recent work in spectral methods includes Specformer (Bo et al., 2023a), combining spectral GNNs with transformer architectures to create learnable set-to-set spectral filters, or the work by (Li et al., 2025) to enhance the scalability of spectral GNNs without decoupling the network architecture, addressing a key limitation in previous approaches. Yang et al. (2024) present a spectral-aware augmentation method that selectively perturbs eigenpairs to preserve task-relevant frequency bands in graph contrastive learning. These advanced spectral methods demonstrate improved performance on various graph learning tasks, but do not specifically target the regression setting or leverage the spectral domain for learning in imbalanced scenarios. Kaya & Çetin Kaya (2024) applied spectral graph convolutional neural networks to Alzheimer's disease diagnosis, showing how these representations can capture complex relationships in biomedical data, but did not address the challenge of imbalanced target distributions.

## 2.3 Graph Sampling and Synthesis in Scientific Domains

Due to domain-specific constraints and validity requirements, scientific applications pose unique challenges for graph-based methods. Yao et al. (2024) provided a comprehensive bibliometric analysis of GNN applications in drug discovery, showing significant growth in this area and highlighting the need for methods to handle the inherent data imbalances in these domains. Similarly, Fan et al. (2024) addressed the challenge of overconfident errors in molecular property classification, demonstrating the importance of uncertainty quantification in imbalanced datasets. These approaches focus primarily on classification rather than regression tasks. On regression tasks, a review on GNNs for predicting synergistic drug combinations (Zhang & Tu, 2023) noted that graph-based models often suffer from imbalanced data distributions, affecting their performance. They emphasized the need for methods to handle such imbalances to improve predictive accuracy effectively.

## 2.4 Molecular Generation

Molecular generation has become a central task in drug discovery, aiming to explore chemical space efficiently while ensuring chemical validity and optimizing for desired properties. Early approaches combined variational autoencoders (VAEs), recurrent neural networks (RNNs), and adversarial models to generate novel chemical structures from latent spaces, as in LatentGAN (Prykhodko et al., 2019), which integrated autoencoding with generative adversarial training for de novo molecular

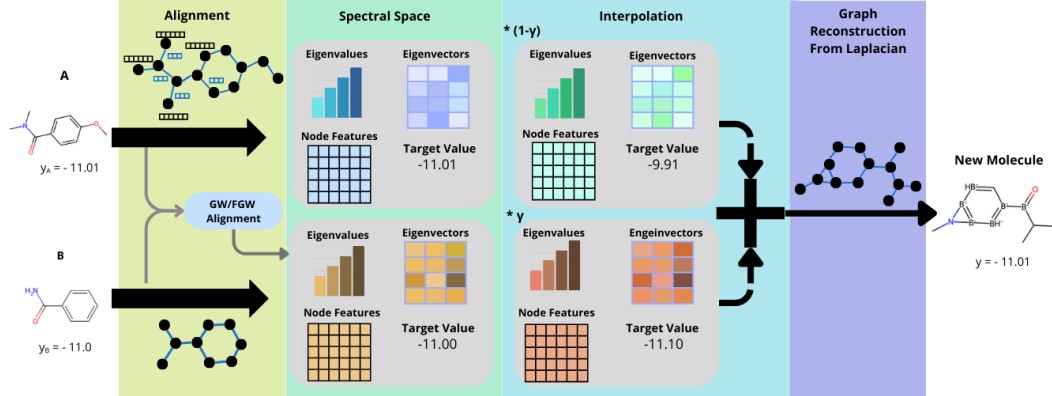

Figure 2: SPECTRA Pipeline. Molecular graphs are first aligned using Gromov–Wasserstein matching, which maps graph B into the structural space of graph A. Their Laplacians are then decomposed into eigenvalues and eigenvectors to obtain a spectral representation. Interpolation is performed in this spectral space-along with node features and target values—by applying a linear combination to each component, producing an interpolated spectral representation. Finally, the resulting Laplacian and node embeddings are used to reconstruct the interpolated molecular graph.

design. More recent methods leverage reinforcement learning to incorporate chemical constraints and multi-objective optimization. For example, DeepGraphMolGen (Khemchandani et al., 2020) employs Graph Convolutional Policy Networks to generate molecules while simultaneously optimizing for drug-likeness and synthetic accessibility, whereas MORLD (Jeon & Kim, 2020) integrates reinforcement learning with docking simulations to propose inhibitors directly guided by protein structures. Conditional generative frameworks, such as MGCVAE (Lee & Min, 2022), enable property-conditioned molecular graph generation, allowing for inverse design tasks like optimizing logP or molar refractivity. Beyond purely graph-based approaches, protein-informed generation methods such as DeepTarget (Chen et al., 2023) directly construct candidate molecules from amino acid sequences of target proteins, bridging structural biology with generative chemistry. Diffusion models have also been widely adopted in molecular generation Zhang et al. (2023), for example, in DiGress Vignac et al. (2022), discrete diffusion process that progressively edits graphs with noise, through the process of adding or removing edges and changing the categories. Despite these advances, most existing models focus on validity, novelty, and property optimization, without explicitly addressing the imbalance of molecular property distributions.

## 2.5 SPECTRA NOVELTY

Our SPECTRA method introduces a spectral-domain augmentation strategy that explicitly targets underrepresented regions of the label space while preserving global graph structure and chemical validity. Unlike many existing approaches that rely on pseudo-labeling or sacrifice accuracy in well-represented regions, SPECTRA generates new, chemically coherent samples where data are sparse, mitigating imbalance without degrading overall performance. By combining spectral alignment with rare-target–aware sampling and validity-preserving reconstruction, it enables interpretable molecule generation and improves regression accuracy in rare but scientifically critical regimes.

## 3 METHOD

We propose a spectral, geometry-aware augmentation and learning pipeline for molecular property prediction that (i) constructs multi-attribute Laplacian representation from molecular graphs; (ii) aligns laplacians and nodes representations of different graphs using (Fused) Gromov–Wasserstein (FGW) couplings; (iii) interpolates eigenvalues and eigenvectors, along to node features in a stable orthonormal basis; and (iv) trains a spectral GNN with edge-aware Chebyshev convolutions on original and augmented samples. Figure 2 summarizes the workflow.

### 3.1 FROM SMILES TO MULTI-ATTRIBUTE GRAPHS

Given a SMILES string $s$, we construct the graph $G = (V, \mathbf{X}, \mathbf{E}, y)$ with RDKit[1]. Where $|V|$ is the set of nodes (atoms), $\mathbf{X} \in \mathbb{R}^{n \times d}$ atom-feature matrix (using OGB utilities)-with $n$ equal the number of nodes and $d$ the dimensionality of node features- and each undirected edge $(u, v) \in E$, for $u, v \in V$, has a 3D attribute vector (bond type, stereo, conjugation). We treat the three edge channels as separate weighted adjacencies $\{\mathbf{W}^{(f)}\}_{f=1}^{F}$ ($F$ is the number of edge-attribute channels) and compute one (unnormalized) Laplacian per channel.

$$\mathbf{L}^{(f)} = \mathbf{D}^{(f)} - \mathbf{W}^{(f)}, \quad \mathbf{D}^{(f)} = \operatorname{diag}(\mathbf{W}^{(f)}\mathbf{1}).$$

Where $\mathbf{1}$ is the vector of ones used for degree computation. To prevent disconnected subgraphs within individual bond-type channels, we add a small constant offset to each corresponding adjacency matrix, ensuring that all nodes maintain minimal connectivity. This adjustment stabilizes the Laplacian spectra across channels and avoids degenerate eigenmodes during spectral interpolation while preserving the relative structural topology. After generating the interpolated Laplacian, the offset is subtracted to restore the original structural scale before reconstruction.

### 3.2 GEOMETRY-AWARE GRAPH ALIGNMENT (FGW)

Network alignment (or graph matching) is a widely studied problem across scientific disciplines. In its basic form, it aims to identify a node correspondence between two graphs that best preserves their structural relationships Lázaro et al. (2025). In molecular applications, this alignment enables the comparison of structural similarity across different compounds Emmert-Streib et al. (2016).

To establish node correspondence between two molecules $A$ and $B$, we solve a Gromov–Wasserstein (GW) or Fused Gromov–Wasserstein (FGW) optimal transport problem on their zero-padded adjacency matrices $\tilde{\mathbf{A}}, \tilde{\mathbf{B}} \in \mathbb{R}^{n \times n}$ (padding each graph to the larger node count). We define probability distributions $p, q \in \Delta^n$ (set of valid probability distributions) over the nodes of $A$ and $B$, respectively. Each entry $p_i$ (or $q_j$) represents the relative "mass" assigned to node $i$ in $A$ (or node $j$ in $B$); in this work we use uniform weights so that $p_i = 1/|V_A|$ and $q_j = 1/|V_B|$. The transport plan $\mathbf{T} \in \mathbb{R}^{n \times n}$ then specifies how this probability mass is moved from each node of $A$ to each node of $B$, effectively giving a soft alignment between their nodes. When node attributes are available, we use FGW with a cost matrix $\mathbf{M}$ that measures feature dissimilarity; otherwise we use pure GW:

$$\mathbf{T}^\star = \arg \min_{\mathbf{T} \in \Pi(p,q)} (1 - \alpha)\, \mathcal{L}_{\text{GW}}(\tilde{\mathbf{A}}, \tilde{\mathbf{B}}, \mathbf{T}) + \alpha \langle \mathbf{M}, \mathbf{T} \rangle,$$

where $\Pi(p, q)$ is the set of couplings with marginals $p$ and $q$, $\mathcal{L}_{\text{GW}}$ is the squared-loss GW discrepancy, and $\alpha \in [0, 1]$ balances structural versus feature similarity. The resulting optimal coupling $\mathbf{T}^\star$ provides a soft node-to-node correspondence; we convert it into a hard one-to-one mapping using the Hungarian assignment on $-\mathbf{T}^\star$ and reorder $\tilde{\mathbf{B}}$ and its features accordingly.

### 3.3 SPECTRAL INTERPOLATION

Given the matched pair, $\mathbf{\Lambda}_A, \mathbf{\Lambda}_B$ being the eigenvalues of A and B, respectively, and $\mathbf{U}_A, \mathbf{U}_B$ being the eigenvectors of A and B, respectively, we diagonalize

$$\mathbf{L}_A = \mathbf{U}_A \mathbf{\Lambda}_A \mathbf{U}_A^\top, \quad \mathbf{L}_B = \mathbf{U}_B \mathbf{\Lambda}_B \mathbf{U}_B^\top,$$

and align eigenvector signs and bases with an orthogonal Procrustes map $\mathbf{R}^\star = \arg \min_{\mathbf{R} \in O(k)} \|\mathbf{U}_A^\top \mathbf{U}_B - \mathbf{R}\|_F$, given $O(k) = \{ R \in \mathbb{R}^{k \times k} : R^\top R = I \}$ where $\| \cdot \|_F$ is the Frobenius norm, and $k$ is the number of eigenvalues/eigenvectors, yielding $\widetilde{\mathbf{U}}_B = \mathbf{U}_B \mathbf{R}^\star$. We then interpolate eigenvalues and bases with a mixing coefficient $\gamma \in (0, 1)$:

$$\mathbf{\Lambda}_\gamma = (1 - \gamma)\mathbf{\Lambda}_A + \gamma \mathbf{\Lambda}_B, \quad \widehat{\mathbf{U}} = (1 - \alpha)\mathbf{U}_A + \gamma \widetilde{\mathbf{U}}_B, \quad \mathbf{U}_\gamma = \operatorname{qr}(\widehat{\mathbf{U}}),$$

where $qr(\cdot)$ factorization used for re-orthogonalization, and synthesize an intermediate Laplacian

$$\mathbf{L}_\gamma = \mathbf{U}_\gamma \mathbf{\Lambda}_\gamma \mathbf{U}_\gamma^\top$$

We repeat this per edge channel ($F$=3).

---

[1]RDKit: https://www.rdkit.org

**Node feature interpolation.** In the matched node domain we perform linear interpolation in the original node space ,where $\widetilde{\mathbf{X}}_B$ is $\mathbf{X}_B$ permuted by the GW/FGW correspondence:

$$\mathbf{X}_\gamma = (1 - \gamma)\,\mathbf{X}_A + \gamma\,\widetilde{\mathbf{X}}_B,$$

### 3.4 Graph Reconstruction from Spectra

For each channel, we map $\mathbf{L}_\gamma^{(f)}$ back to a nonnegative adjacency:

$$\mathbf{W}_\gamma^{(f)} = \max\!\big(0,\; -\mathbf{L}_\gamma^{(f)} + \operatorname{diag}(\mathbf{L}_\gamma^{(f)})\big), \quad \operatorname{diag}(\mathbf{W}_\gamma^{(f)}) = \mathbf{0}.$$

We remove degrees and clip negatives.We then assemble multi-attribute edges by scanning $(u, v)$ with any positive channel weight and stacking per-channel features. The scalar label is interpolated as $y_\alpha = (1 - \gamma)y_A + \gamma y_B$.

### 3.5 Rarity-Aware Pair Selection and Augmentation Budget

We compute a KDE over training labels to estimate density $\rho(y)$ and define rarity weights $w_i \propto 1/\rho(y_i)$ (normalized). Each training molecule $i$ receives an augmentation budget $\lfloor w_i \cdot N \cdot \texttt{perc} \rfloor$ ($\texttt{perc} \in [0, 1]$ is a global rate). For molecule $i$, we sort neighbors by $|y_i - y_j|$ and generate pairs $(i, j)$ in that order, producing up to the allocated number of augmented graphs.

### 3.6 Graph Validity and Conversion Back to Molecules

To verify that the augmented graphs correspond to real chemical compounds, we convert each generated graph back into a SMILES string and validate its chemical consistency with RDKit. Each node's feature vector is translated back into an atom description by mapping the numerical features to basic chemical properties: the atom type (e.g., carbon, oxygen), its charge, chirality, hybridization state, and whether it is aromatic. When a property cannot be confidently inferred, we assign a safe default (treating it as carbon, for example). Bonds are reconstructed from edge attributes, including type, stereochemistry, and conjugation, while avoiding duplicates to maintain a valid simple graph.

The resulting editable molecule is then sanitized with RDKit to enforce valence rules, aromaticity perception, and proper connectivity. If strict sanitization fails, a relaxed mode attempts to correct hydrogen counts and minor inconsistencies. Finally, the molecule is converted to a canonical SMILES string; graphs that cannot be sanitized are marked invalid.

### 3.7 Spectral GNN with Edge-Aware Chebyshev Convolutions

We use a stack of $L$ spectral graph-convolutional blocks based on Chebyshev filters (`ChebConv`; Defferrard et al., 2016), batch normalization, SiLU activation, and dropout. Let $\mathbf{H}^{(0)} = \mathbf{X}$ denote the input node features, and $\mathbf{H}^{(\ell)} \in \mathbb{R}^{n \times d_\ell}$ the hidden representation at layer $\ell$. Given a graph operator $\mathbf{A}$ (constructed from the multi-channel Laplacians) and learnable edge-projection parameters $\mathbf{w}_e$, each block computes

$$\mathbf{H}^{(\ell+1)} = \operatorname{Drop}\!\Big(\operatorname{SiLU}\!\Big(\operatorname{BN}\big(\operatorname{ChebConv}_K(\mathbf{H}^{(\ell)},\, \mathbf{A},\, \mathbf{w}_e)\big)\Big)\Big),$$

where $K$ is the Chebyshev polynomial order and $\operatorname{Drop}(\cdot)$ is applied with probability $p_{\text{drop}}$. Each edge $(u, v)$ carries a 3-dimensional attribute vector $\mathbf{e}_{uv} \in \mathbb{R}^3$ (bond type, stereochemistry, conjugation), which is projected through $\mathbf{w}_e$ and incorporated into the convolutional kernels.

## 4 Results

We evaluate our methods across three benchmark datasets (FreeSolv, ESOL, Lipo and QM9, in detail in Appendix A.1. To assess the model's ability to generate a diverse set of real molecules distinct from the training data (RQ1) we evaluated the generated drugs against Lipinski's Rule of Five Giménez et al. (2010) and assessed other important properties: quantitative estimate of drug-likeness (QED) (Bickerton et al., 2012) and synthetic accessibility score (SA) (Ertl & Schuffenhauer, 2009). We also use standard metrics like validity, uniqueness, novelty.

We also evaluate the impact of the different steps of our methods along with their different parameters(RQ2), assess the predictive performance of our model against state-of-the-art methods (RQ3), and analyze its behavior across the entire target domain to understand improvements in low-density regions compared to high-density regions (RQ4). We also assess the computational efficiency of our approach relative to existing methods (RQ5).

## 4.1 MOLECULE GENERATION QUALITY (RQ1)

To evaluate the quality of the generated molecules, we first assess their *validity*, *uniqueness*, and *novelty*. Validity is the fraction of generated molecules that are chemically valid, uniqueness is the fraction of valid molecules that are non-duplicate, and novelty is the fraction of unique molecules not present in the training set (Flam-Shepherd et al., 2022). Table 1 shows that all generated molecules are chemically valid, which is expected since the generated structures are passed through a validity-checking step. Also, achieve high uniqueness and novelty. Uniqueness is slightly lower for FreeSolv, likely due to its smaller chemical space, but novelty is the highest value (1.0), while QM9 presents the highest uniqueness, indicating that our augmentation strategy produces new, valid structures rather than replicating training molecules.

To further understand the impact on chemical space, we visualize original and augmented molecules using t-SNE on Morgan fingerprints (Figure 3). Augmented molecules populate sparse regions, improving coverage and mitigating distributional imbalance. This broader coverage helps reduce bias toward overrepresented regions and supports better generalization to underrepresented subspaces.

| Dataset | | Validity | Unique | Novelty | Atoms | | | Rings | | | Bonds | | |
|---|---|---|---|---|---|---|---|---|---|---|---|---|---|
| | | | | | *min* | *mean* | *max* | *min* | *mean* | *max* | *min* | *mean* | *max* |
| FreeSolv | Orig | | | | 1 | 8.73 | 24 | 0 | 0.66 | 5 | 0 | 8.39 | 25 |
| | Gen | 1.000 | 0.568 | 1.000 | 5 | 12.06 | 20 | 0 | 1.77 | 8 | 4 | 12.70 | 23 |
| ESOL | Orig | | | | 1 | 13.28 | 55 | 0 | 1.39 | 8 | 0 | 13.67 | 62 |
| | Gen | 1.000 | 0.661 | 0.949 | 2 | 19.83 | 28 | 0 | 2.86 | 7 | 1 | 21.69 | 31 |
| Lipo | Orig | | | | 7 | 27.04 | 115 | 0 | 3.49 | 13 | 7 | 29.50 | 118 |
| | Gen | 1.000 | 0.706 | 0.992 | 10 | 22.80 | 115 | 0 | 3.25 | 9 | 10 | 24.84 | 119 |
| QM9 | Orig | | | | 1 | 8.80 | 9 | 0 | 1.74 | 8 | 0 | 9.40 | 13 |
| | Gen | 1.000 | 0.928 | 0.819 | 4 | 8.84 | 9 | 0 | 1.86 | 8 | 3 | 9.54 | 13 |

Table 1: Validity, uniqueness, and novelty of generated molecules along with atom, ring and bonds statistics for Original vs. Generated molecules across datasets.

We also assess structural complexity by comparing atom, ring, and bond statistics between original and augmented molecules (Table 1). In FreeSolv and ESOL, augmented molecules are generally larger and more cyclic, with higher mean atom, ring, and bond counts, while Lipo and QM9 show similar overall distributions. Figure 7 (Appendix) further shows that augmentation increases atom-type diversity and shifts ring and bond-type compositions. QM9 is the only exception, exhibiting nearly unchanged distributions due to its size-constrained and chemically homogeneous design. Overall, these results indicate that our method broadens scaffold diversity while remaining within chemically reasonable bounds, reinforcing the improved dataset coverage and property–target alignment.

Furthermore, we evaluated the generated drugs against Lipinski's Rule of Five, a widely used guideline in drug discovery for assessing physicochemical properties essential for oral bioavailability Giménez et al. (2010). These rules include drugs having five or fewer hydrogen bond donors (HBD), a molecular weight (MW) of less than 500 Da, a partition coefficient (logP) Wildman & Crippen (1999) of less than five, and ten or fewer hydrogen bond acceptors (HDA). As can be seen in Figure 4 that most of the generated molecules satisfy Lipinski's Rule of Five. Also, it can be seen at Figure 8, in Appendix A.2, that the generated drugs have an acceptable average distribution of QED and SAS. This indicates that, despite variability, the generative process is capable of producing molecules that maintain desirable medicinal chemistry characteristics.

## 4.2 ABALATION STUDY AND PARAMETER ANALYSIS (RQ2)

To disentangle the contributions of each design choice, we perform an ablation study on interpolation (Inter), spectral alignment (FGW), and KDE-based augmentation (Table 2). Removing interpolation

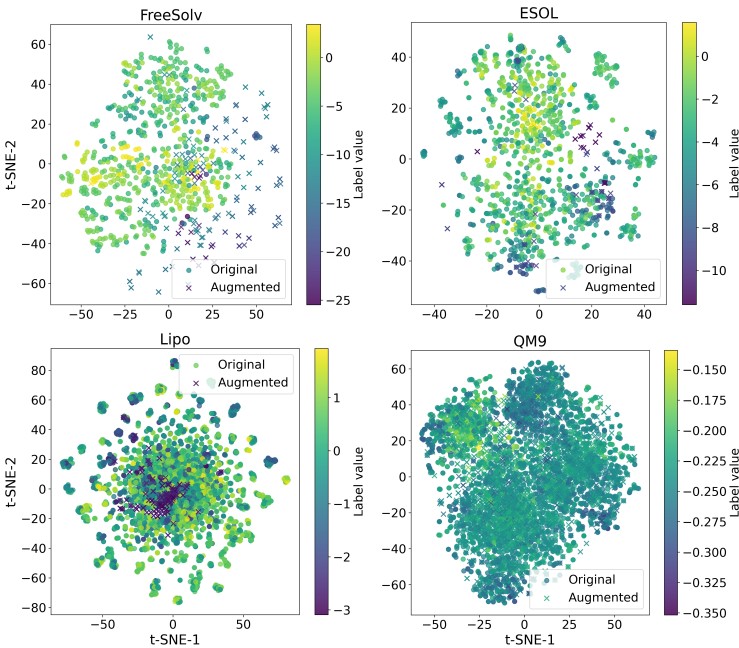

Figure 3: **t-SNE visualization of Morgan fingerprints** comparing original and augmented samples for each dataset. QM9 plot presents 2.5% of the generated and original dataset sampled randomly for visualization purposes.

and simply copying samples results in a consistent drop in performance across all datasets, indicating that sample diversity is essential. Excluding FGW alignment also reduces performance–except on ESOL–highlighting the importance of geometry-aware alignment for structurally diverse molecules. Likewise, removing KDE-based augmentation degrades performance in imbalanced regions, underscoring the value of density-aware augmentation for improved generalization. Overall, the full model achieves the best or near-best performance across all datasets. Figure 9 in Appendix A.3 shows the effect of the alignment weight $\alpha$ and mixing coefficient $\gamma$, with the best results obtained for $0.25 \leq \alpha \leq 0.75$ and $\gamma \in 0.1, 0.5$. This demonstrates that balanced structural alignment and appropriate mixing jointly contribute to performance gains, confirming that each component of the pipeline plays a distinct and influential role.

| Aug | Inter | Align | KDE | ESOL | FreeSolv | Lipo | QM9 |
|-----|-------|-------|-----|------|----------|------|-----|
| ✗ | ✗ | ✗ | ✗ | 0.586(0.568) [87.84s] | 0.926(1.125) [43.03s] | 0.408(0.384) [334.27s] | 0.00369(0.005) [6980.84s] |
| ✓ | ✗ | ✗ | ✗ | 0.542(0.516) [57.98s] | 0.799(1.087) [48.04s] | 0.383(0.361) [479.52s] | 0.00360(0.005) [7726.88s] |
| ✓ | ✗ | ✗ | ✓ | 0.542(0.516) [75.43s] | 0.799(1.087) [52.18s] | 0.383(0.361) [355.64s] | 0.00360(0.005) [7292.71s] |
| ✓ | ✓ | ✗ | ✗ | 0.542(0.512) [75.58s] | 0.863(1.128) [58.28s] | 0.384(0.371) [372.92s] | 0.00369(0.005) [7109.36s] |
| ✓ | ✓ | ✗ | ✓ | **0.534(0.515) [84.42s]** | 0.869(1.113) [57.26s] | 0.378(0.362) [380.28s] | 0.00374(0.005) [7046.01s] |
| ✓ | ✓ | ✓ | ✗ | 0.571(0.557) [95.20s] | 0.807(1.164) [55.44s] | 0.389(0.363) [679.49s] | 0.00370(0.005) [8323.45s] |
| ✓ | ✓ | ✓ | ✓ | **0.534(0.525) [95.90s]** | **0.769(1.023) [113.99s]** | **0.377(0.359) [691.24s]** | **0.00357(0.005) [11582.16s]** |

Table 2: Ablation study with incremental addition of augmentation (Aug), alignment (Align), and KDE prior. Results are reported as $mean(std)_{[MeanRunTime]}$ of per-sample errors over multiple runs. Best results per dataset are highlighted in bold.

### 4.3 PREDICTIVE PERFORMANCE (RQ3)

We compare our proposed method (**SPECTRA**) against representative state-of-the-art molecular representation learning models, including contrastive methods (GraphCL(You et al., 2020), Mol-CLR (Wang et al., 2022)), language–graph hybrids (Molformer Ross et al. (2022), Chemb Defferrard et al. (2016)), and recent GNN-based frameworks (HiMol (Zang et al., 2023), SGIR (Liu et al., 2023a)). Table 3 reports both the general prediction accuracy, measured by mean absolute error (MAE), and performance under imbalance, measured by squared error relevance area (SERA).

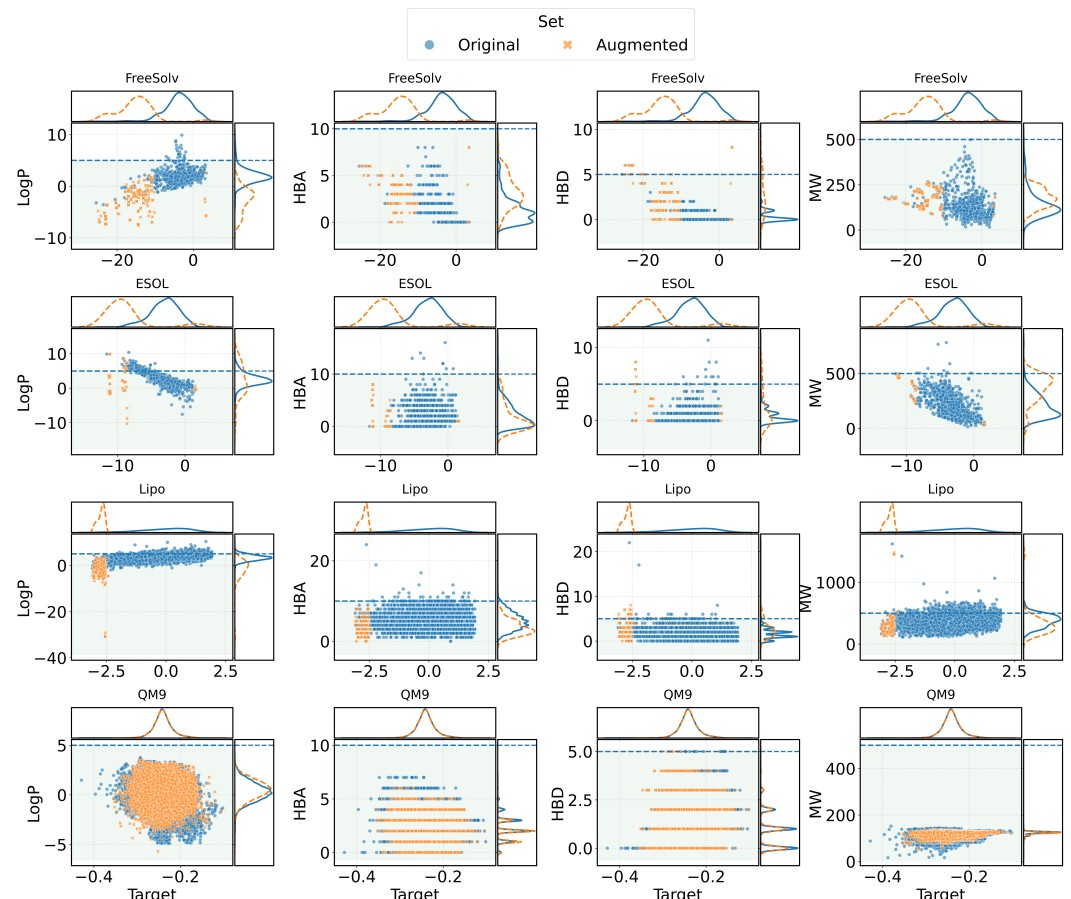

Figure 4: Joint distributions of target property values versus key physicochemical descriptors (LogP, HBA, HBD, and molecular weight) for both original (blue circles) and augmented molecules (orange crosses) across datasets. The shaded regions highlight Lipinski-compliant intervals, including $LogP \leq 5$, $MW \leq 500$ Da, $HBD \leq 5$ and $HBA \leq 10$ and acceptors (HBA).

Table 5, in Appendix, reports pairwise statistical tests demonstrating that SPECTRA significantly outperforms competing models on most datasets, with only a few isolated cases where differences are negligible. The experimental setup is presented in Appendix A.5.

| Model | MAE (mean ± var) | | | | SERA (mean ± var) | | | |
|---|---|---|---|---|---|---|---|---|
| | ESOL | FreeSolv | Lipo | QM9 | ESOL | FreeSolv | Lipo | QM9 |
| Chemb | 0.59 ± 0.32 | 0.93 ± 1.26 | 0.41 ± 0.15 | 0.00360 ± 0.0000 | 0.25 ± 0.01 | 1.07 ± 0.19 | 0.11 ± 0.00 | **0.00002 ± 0.0000** |
| GraphCL | 0.78 ± 0.40 | 1.76 ± 2.30 | 0.73 ± 0.30 | 0.01077 ± 0.0001 | 0.36 ± 0.00 | 2.59 ± 0.45 | 0.40 ± 0.00 | 0.00011 ± 0.0000 |
| HiMol | 0.51 ± 0.22 | 0.97 ± 1.46 | 0.41 ± 0.14 | 0.06296 ± 0.0018 | 0.17 ± 0.00 | 1.34 ± 0.74 | 0.10 ± 0.00 | 0.00183 ± 0.0000 |
| MolCLR | 0.73 ± 0.40 | 1.12 ± 1.31 | 0.43 ± 0.14 | 0.00364 ± 0.0000 | 0.32 ± 0.00 | 1.36 ± 0.18 | 0.11 ± 0.00 | **0.00002 ± 0.0000** |
| Molformer | 1.66 ± 1.68 | 2.84 ± 6.87 | 0.81 ± 0.38 | 0.01628 ± 0.0002 | 2.77 ± 0.01 | 10.61 ± 12.16 | 0.54 ± 0.00 | 0.00038 ± 0.0000 |
| SGIR | **0.46 ± 0.19** | **0.68 ± 0.85** | **0.37 ± 0.13** | **0.00341 ± 0.0000** | **0.13 ± 0.00** | **0.69 ± 0.05** | **0.09 ± 0.00** | **0.00002 ± 0.0000** |
| SPECTRA | 0.53 ± 0.28 | 0.77 ± 1.05 | 0.38 ± 0.13 | 0.00357 ± 0.0000 | 0.20 ± 0.00 | 0.95 ± 0.29 | **0.09 ± 0.00** | **0.00002 ± 0.0000** |

Table 3: Mean absolute error (MAE) and SERA with variance for each model across four datasets. Lower values indicate better performance. Bold models are the best results while underlined denote second best.

SPECTRA achieves consistently strong performance across all datasets. On ESOL, Lipo and QM9, SPECTRA is competitive with the best-performing models, and on FreeSolv, it surpasses most baselines with the second-best overall performance. While SGIR attains the best MAE and SERA scores on average, SPECTRA achieves a stable balance across datasets and excels in capturing underrepresented regions, as reflected in its low SERA values. This indicates that our augmentation and

spectral alignment strategies effectively improve prediction in imbalanced regimes without sacrificing overall performance.

## 4.4 ERROR DISTRIBUTION ACROSS TARGET RANGES (RQ4)

Figure 5 further dissects MAE by target value ranges. We observe that baseline models often suffer from considerably higher errors in the low-density regions, consistent with the imbalance in the training data. In contrast, SPECTRA demonstrates markedly lower errors in these sparse regions, highlighting its strength in addressing imbalance.

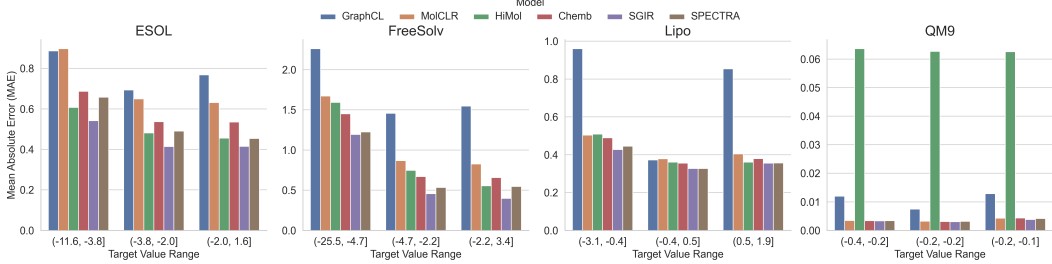

Figure 5: **Mean Absolute Error (MAE) distribution** across target value ranges for each dataset. Colors correspond to different models as indicated in the legend.

## 4.5 TIME EFFICIENCY ANALYSIS (RQ5)

Besides predictive accuracy, computational efficiency is also crucial for real-world deployment. Figure 6 shows the runtime distribution of all models across the three datasets. SPECTRA consistently ranks among the fastest methods, running substantially quicker than Molformer and SGIR (2-5X faster), and demonstrating clear time advantages over the baseline models.

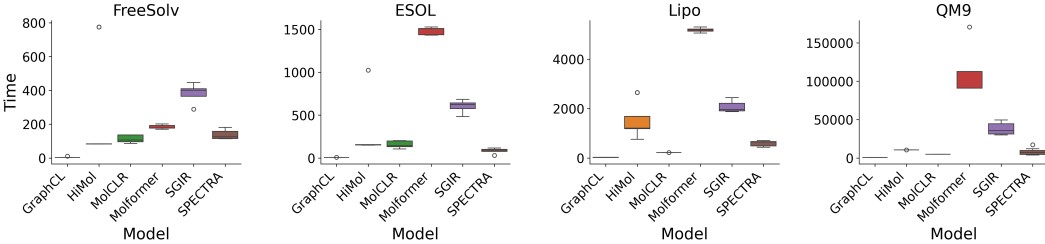

Figure 6: Boxplots compare the inference/training runtime of the models on datasets. Each subplot corresponds to one dataset, and box distributions summarize the variability of runtime on train across 5 folds cross-validation.

## 5 CONCLUSION

Experiments across benchmark datasets show that our method improves predictive accuracy in rare but critical regimes, preserves property–target correlations, and achieves a favorable balance between accuracy and efficiency. These results establish spectral augmentation as a promising and interpretable strategy for tackling imbalance in molecular property prediction and related graph-structured scientific domains. Future work will explore extending the framework to multi-property prediction, incorporating additional modalities such as 3D features, and developing more efficient variants of the FGW alignment and spectral decomposition steps to reduce computational complexity and improve runtime. We also plan to investigate hybrid strategies that combine spectral augmentation with model-based pseudo-labeling to further enhance flexibility and performance.

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

# A  APPENDIX

## A.1  DATASET DETAILS

Our experimental evaluation uses molecular regression tasks from MoleculeNet (Wu et al., 2018), specifically ESOL, FreeSolv, Lipophilicity (Lipo) and QM9. A brief summary of these datasets is provided in Table 4. The QM9 dataset provides 12 quantum-chemical properties, from which we selected the HOMO energy as the target for evaluating our method.

Table 4: Summary of Molecular Property Datasets

| Dataset | # of Compounds | Description |
|---|---|---|
| ESOL | 1,128 | Water solubility (log solubility in mol/L) |
| FreeSolv | 642 | Hydration free energy in water |
| Lipophilicity | 4,200 | Octanol/water distribution coefficient (logD at pH 7.4) |
| QM9 | 133,886 | Geometric, energetic, electronic and thermodynamic properties of DFT-modelled small molecules. |

## A.2  MOLECULAR PROPERTIES ANALYSIS

Figure 7 compares the global structural composition of molecules in the original datasets with those produced by our augmentation method. Overall, the augmented molecules exhibit greater variability in atom types—most notably an increased proportion of "Other" atoms compared to the original sets. Ring compositions also shift, with the ESOL augmented set showing a higher fraction of six-membered (hexagonal) rings. Bond-type distributions remain broadly similar, although subtle changes appear: aromatic bonds increase in the generated ESOL molecules, whereas single bonds become more prevalent in the Lipo augmented set. QM9 shows nearly identical structural distributions. This is can be explanided by its inherent size-constrained and chemically homogeneous nature due to the strict rules used in its construction.

As shown in Figure 8, the joint distributions of SA and QED demonstrate that the generated molecules generally fall within acceptable ranges for both properties. While some augmented samples exhibit higher SA or lower QED—suggesting increased synthetic complexity and reduced drug-likeness—the overall distribution still contains many promising candidates with low SA and high QED Ertl & Schuffenhauer (2009); Li et al. (2024).

## A.3  PARAMETER EVALUATION

Figure 9 presents the impact in validaiton sey of different values of gammas and alphas, exploring the joint effects of the alignment parameter $\alpha$ (which controls the balance of structural features) and the mixing coefficient $\gamma$. The heatmap summarizes the corresponding mean absolute error (MAE) across combinations of $(\alpha, \gamma)$, allowing visual inspection of how these parameters interact and influence predictive performance. We choose lower values for $\gamma$ because we want to be closer to the sample chosen for augmentation.

## A.4  STATISTICAL IMPROVEMENT

Table 5 provides a detailed statistical comparison between SPECTRA and all baseline models across the ESOL, FreeSolv, Lipo , and QM9 datasets. For each pairwise comparison, we compute the raw absolute-error improvement, defined as the per-sample difference in absolute prediction errors between SPECTRA and the competing model. Positive values indicate that SPECTRA produces lower predictive error. To assess whether these differences are statistically meaningful, we apply the paired t-test. The "Best Model" column identifies which method achieves the lowest absolute errors on average, while the "Significant" column reports whether the observed improvement is statistically significant under a threshold of $\alpha = 0.05$.

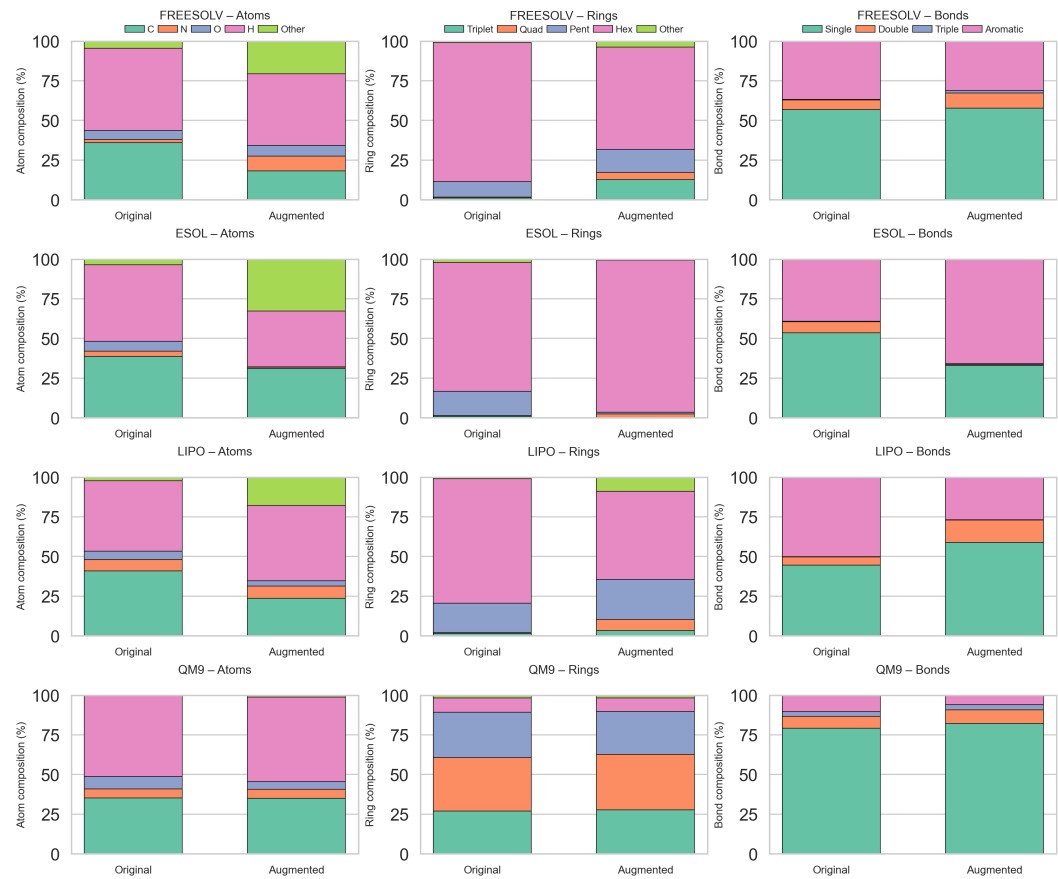

Figure 7: Comparison of the structural features between the original drugs from the FreeSolv, ESOL, Lipo and QM9 datasets and the drugs generated by SPECTRA. The first chart represents the percentage of different atoms (C=carbon, H=Hidrogen, N=nitrogen, O=oxygen, Other), the second chart shows the proportion of different ring types (Tri, Quad, Pent, Hex, Other), while the third chart reflects the diversities of bond types (Single, Double, Triple) in the molecules.

Overall, the results demonstrate that SPECTRA consistently outperforms most baselines—with particularly strong margins on FreeSolv—showing both large error reductions and highly significant Wilcoxon $p$-values. A few cases, such as HiMol and SGIR on specific datasets, show either negligible differences or slight advantages for the baseline, and these are reflected accordingly in the statistical tests.

## A.5 EXPERIMENTAL SETUP

All experiments were conducted on a Linux server equipped with two 12-core `Intel(R) Haswell` processors, 256 GB of RAM, and four `NVIDIA A100` GPUs, each with 80 GB of memory. Our method is implemented in `Python 3.8.19` using `PyTorch 2.1.2`. We used Chebyshev GCN (cheb) (Defferrard et al., 2016). We perform a manual hyperparameter search over the following ranges:

- **Hidden dimension:** $\{128, 256, 512\}$

- **Number of layers:** $\{3, 4, 5\}$

- **Dropout:** $\{0.0, 0.1, 0.3\}$

- **Learning rate:** $\{10^{-3}, 2 \times 10^{-3}, 5 \times 10^{-4}\}$

- **Chebyshev filter order ($k$):** $\{2, 3, 5\}$

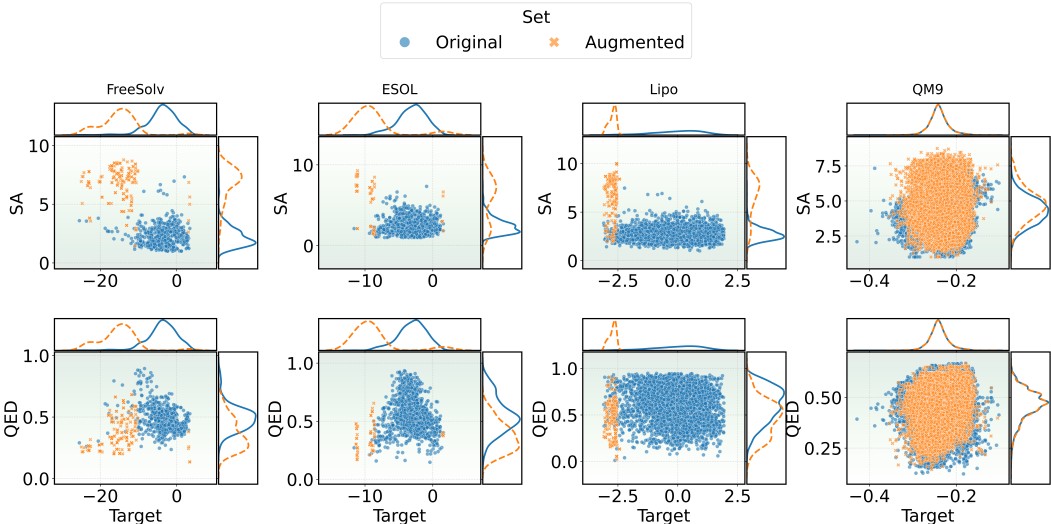

Figure 8: Joint distributions of target property values versus SA and QED scores for both original and augmented molecules across the FreeSolv, ESOL, and Lipo datasets. The faded green regions indicate favorable ranges for each property—lower values for synthetic accessibility (SA), corresponding to easier synthesis, and higher values for QED, reflecting greater drug-likeness.

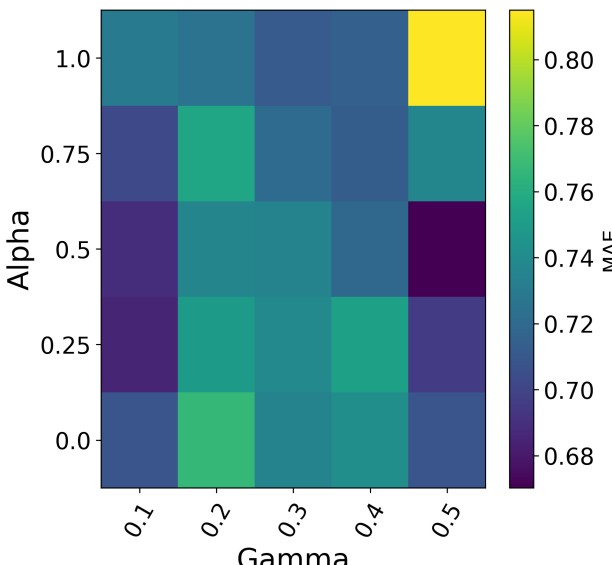

Figure 9: MAE surfaces for the $\alpha$ and $\gamma$ hyperparameter in the FreeSolv dataset. Each heatmap shows the MAE computed on the validation set, with $\alpha$ balancing structural features for alignment and $\gamma$ as the mixing coefficient.

- **Epochs:** $\{500\}$

- **Batch size:** $\{32, 64\}$

- **Gamma:** $\{0.1, 0.2, 0.3, 0.4, 0.5\}$

- **Alpha:** $\{0.1, 0.25, 0.5, 0.75, 1\}$

- **Augmentation Percentage(%):** $\{0.1, 0.15, 0.2, 0.25\}$

|    | Dataset  | Model     | t-test | Mean Improvement | Best Model | Significant |
|----|----------|-----------|--------|------------------|------------|-------------|
| 0  | esol     | Chemb     | 0.000  | 0.052            | SPECTRA    | Yes         |
| 1  | esol     | GraphCL   | 0.000  | 0.248            | SPECTRA    | Yes         |
| 2  | esol     | HiMol     | 0.212  | -0.019           | HiMol      | No          |
| 3  | esol     | MolCLR    | 0.000  | 0.192            | SPECTRA    | Yes         |
| 4  | esol     | Molformer | 0.000  | 1.124            | SPECTRA    | Yes         |
| 5  | esol     | SGIR      | 0.000  | -0.077           | SGIR       | Yes         |
| 6  | freesolv | Chemb     | 0.000  | 0.158            | SPECTRA    | Yes         |
| 7  | freesolv | GraphCL   | 0.000  | 0.987            | SPECTRA    | Yes         |
| 8  | freesolv | HiMol     | 0.000  | 0.197            | SPECTRA    | Yes         |
| 9  | freesolv | MolCLR    | 0.000  | 0.353            | SPECTRA    | Yes         |
| 10 | freesolv | Molformer | 0.000  | 2.074            | SPECTRA    | Yes         |
| 11 | freesolv | SGIR      | 0.035  | -0.086           | SGIR       | Yes         |
| 12 | qm9      | Chemb     | 0.000  | 0.000            | SPECTRA    | Yes         |
| 13 | qm9      | GraphCL   | 0.000  | 0.007            | SPECTRA    | Yes         |
| 14 | qm9      | HiMol     | 0.000  | 0.059            | SPECTRA    | Yes         |
| 15 | qm9      | MolCLR    | 0.000  | 0.000            | SPECTRA    | Yes         |
| 16 | qm9      | Molformer | 0.000  | 0.013            | SPECTRA    | Yes         |
| 17 | qm9      | SGIR      | 0.000  | -0.000           | SGIR       | Yes         |
| 18 | lipo     | Chemb     | 0.000  | 0.032            | SPECTRA    | Yes         |
| 19 | lipo     | GraphCL   | 0.000  | 0.353            | SPECTRA    | Yes         |
| 20 | lipo     | HiMol     | 0.000  | 0.033            | SPECTRA    | Yes         |
| 21 | lipo     | MolCLR    | 0.000  | 0.052            | SPECTRA    | Yes         |
| 22 | lipo     | Molformer | 0.000  | 0.431            | SPECTRA    | Yes         |
| 23 | lipo     | SGIR      | 0.153  | -0.007           | SGIR       | No          |

Table 5: Statistical comparison between SPECTRA and baseline models across all datasets. The table reports raw absolute-error improvements (positive values indicate lower error for SPECTRA), T-test p-values, the best-performing model for each dataset comparison, and whether the difference is statistically significant at $\alpha = 0.05$.

The optimal configuration uses a hidden dimension of 128, four layers, a dropout rate of 0.1, a learning rate of $2 \times 10^{-3}$, and hyperparameters $k = 2$, $\alpha = 0.5$, and $\gamma = 0.5$. The selected augmentation rates are 0.25 for FreeSolv and 0.10 for ESOL, Lipo and QM9.

Our code and data are available on GitHub [2].

## B  THE USE OF LARGE LANGUAGE MODELS (LLMS)

LLM was used just to polish grammar.

---

[2]https://anonymous.4open.science/r/SPECTRA-0D3C

