# OpenReview forum: "SPECTRA: Spectral Target-Aware Graph Augmentation for Imbalanced Molecular Property Regression"
_ICLR.cc/2026/Conference — ICLR 2026 Conference Desk Rejected Submission_

### Official Review · Reviewer_pT5c · 2025-10-30

**Soundness:** 3
**Presentation:** 3
**Contribution:** 2
**Rating:** 4
**Confidence:** 3

**Summary:**

The paper introduces SPECTRA, a spectral-domain augmentation framework for imbalanced molecular property regression. It constructs multi-attribute Laplacian representations of molecular graphs, aligns pairs of molecules using Gromov–Wasserstein matching, and interpolates their eigenvalues, eigenvectors, and node features in a shared spectral basis. From these interpolated spectra, new molecular graphs are reconstructed, then validated and converted back into SMILES. A rarity-aware sampling scheme ensures that augmentation is concentrated in low-density regions of the target distribution. Finally, the augmented data are used to train a spectral GNN with edge-aware Chebyshev convolutions. The approach yields good, yet not SOTA results on 3 benchmarks.

**Strengths:**

- The authors address a gap in the literature—imbalanced regression. This is a relevant and potentially impactful contribution, particularly for low-data regimes in chemistry.

- The manuscript is clearly written and well organized.

- The model is extensively ablated, and the results are presented with clarity in dedicated figures. While performance is generally solid, SPECTRA does underperform one baseline (SGIR).

**Weaknesses:**

- The related work overlooks the field of graph diffusion models, which have set the state of the art since DiGress (2022).

- The motivation for the proposed approach is unconvincing: why is graph matching required prior to interpolation?

- There is no guarantee that interpolation in graph space yields molecules with the desired properties in low-density regions. In other words, interpolating graphs does not necessarily correspond to meaningful interpolation in chemical space. It doesn't even guarantee that the graphs coresponds to valid molecules, as demonstrated by the ad-hoc procedure used to convert those graphs into valid samples.

- Overall, Figure 2 doesn't provide much intuition on the approach.

- SGIR consistently outperforms SPECTRA across all evaluated aspects.

**Questions:**

- Could you please elaborate the latter part of this sentence : " Each node’s feature vector is decoded into an atom specification (atomic number, charge, chirality, hybridization, aromaticity), falling back to reasonable defaults when attributes are missing. " (l 280-281)

- You mention that you treat each edge channel as an adjacency matrix and compute the Laplacian. Therefore, it's likely that each of the corresponding graphs will be disconnected and scattered into small subgraphs, especially for double and triple bonds. Is computing the Laplacian relevant in that case ?

- cf weaknesses, perhaps I missed something but it's unclear to me why you need graph matching.

---

> ### Author Response · Authors · 2025-11-21
>
> We appreciate the reviewer’s thoughtful feedback, which was extremely valuable in helping us refine and strengthen the manuscript.
>
> **Weaknesses**
> * We agree that the related work section initially lacked discussion of diffusion-based generative models, and we have now incorporated this.
>
> * We also clarified the description of network alignment in the methods section. In its basic form, network alignment (or graph matching) seeks a correspondence between nodes in two graphs that best preserves their structural relationships [1]. In chemistry, this is commonly used to assess structural similarity across molecules [2]. Without graph alignment, the node indices of the two graphs do not correspond, causing their eigenvectors to represent unrelated structural regions. Interpolating unaligned spectra produces Laplacians that no longer reflect meaningful graph structure, leading to invalid or chemically inconsistent molecules after reconstruction. Alignment is therefore essential to ensure that interpolation operates on comparable structural components.
>
> * Regarding the assumption about valence or structural consistency, we acknowledge that there is no formal guarantee; however, this heuristic is widely used in molecular generation. Many generative frameworks produce very large sets of candidate molecules and subsequently filter or rank them based on synthesizability, stability, or task-specific criteria. For example, GenH [3] explores a synthesizable space exceeding 106010^{60} molecules by applying 115 reactions to 223,244 building blocks, using additional classification and denoising steps to select valid reactants during generation. Likewise, diffusion models such as GeoDiff [4] generate extensive molecular candidates that are later evaluated or scored. While not theoretically guaranteed, this assumption is consistently adopted in generative chemistry pipelines and has been effective in practice.
>
> * We have updated Figure 2 for clarity with a clearer diagram-like illustration of the spectral interpolation process, removing unnecessary formulas.
>
> * Although SGIR shows comparable predictive performance, our method achieves substantially better runtime (2-5X faster), with a runtime comparable to more time-efficient methods. This updated visualization more effectively shows that our method achieves competitive runtime and offers a favorable balance between predictive accuracy and computational efficiency (Presentation/Soundness), which we have now highlighted more clearly in the revised manuscript (Contribution).
>
> [1] Lázaro, T., Guimerà, R. & Sales-Pardo, M. Probabilistic alignment of multiple networks. Nat Commun 16, 3949 (2025). https://doi.org/10.1038/s41467-025-59077-7
> [2] Emmert-Streib, F., Dehmer, M. & Shi, Y. Fifty years of graph matching, network alignment and network comparison. Inf. Sci. 346-347, 180–197 (2016).
> [3] Gao, Wenhao, Shitong Luo, and Connor W. Coley. "Generative artificial intelligence for navigating synthesizable chemical space." arXiv preprint arXiv:2410.03494 (2024).
> [4] Xu, Minkai, Lantao Yu, Yang Song, Chence Shi, Stefano Ermon, and Jian Tang. "Geodiff: A geometric diffusion model for molecular conformation generation." arXiv preprint arXiv:2203.02923 (2022).
>
> **Questions:**
>
> 1. We updated our methods section with the following sentence: “Each node’s feature vector is translated back into an atom description by mapping the numerical features to basic chemical properties: the atom type (e.g., carbon, oxygen), its charge, chirality, hybridization state, and whether it is aromatic. When a property cannot be confidently inferred, we assign a safe default (such as treating the atom as carbon).”
>
> 2. This is an aspect we implemented but inadvertently omitted from the final manuscript (now added to Section 3.1 of the revised paper). The solution we use is as follows: to prevent disconnected subgraphs within individual bond-type channels, we add a small constant offset to each corresponding adjacency matrix, ensuring that all nodes retain minimal connectivity. This stabilizes the Laplacian spectra across channels, avoids degenerate eigenmodes during spectral interpolation, and preserves the relative structural topology. After generating the interpolated Laplacian, we subtract the offset to restore the original structural scale before reconstruction.
>
> 3. Responded to the first weakness.

---

> > ### Author Response · Authors · 2025-11-26
> > **New results**
> >
> > **Contribution**. We have finalized new experiments on the large QM9 dataset (133,885 molecules), further reinforcing the soundness and scalability of our approach:
> > * In terms of predictive performance, our method remains significantly better in predictive performance w.r.t. existing baselines except SGIR.
> > * In relation to SGIR, we still obtain comparable performance, but our approach offers a substantial efficiency advantage, achieving an average 4X faster runtime and providing a better balance between performance and computational efficiency.
> > * Also, on the significance of our method generating chemically plausible and potentially applicable in real-world discovery settings – another distinctive factor w.r.t. SGIR – the generated samples by SPECTRA in this new dataset still fall within Lipinski-compliant ranges, exhibit high uniqueness (0.928) and novelty (0.819), while maintaining structural complexity distributions that closely match those of the original molecules.

---

> > ### Author Response · Authors · 2025-11-26
> > **Presentation Improvements**
> >
> > **Presentation**: We acknowledge the presentation issues raised by the reviewers and have revised the paper accordingly.
> > * We updated the main figure (Figure 2) by removing equations and redesigning it as a cleaner, more diagrammatic illustration that clearly highlights each step of our approach.
> > * Figure 1 was also improved by adding a dataset example and presenting the distributions more clearly.
> > * We consolidated the original tables into a single, clearer table, and in Figure 3, we now color the samples by their target values to better illustrate how the augmented space compares to the original distribution.
> > * We also expanded and refined the ablation table to show performance differences across a broader set of methodological combinations, including augmentation, interpolation, alignment, and the KDE-based inverse distribution.
> > * Finally, we corrected missing symbol explanations in the methodology section, fixed minor errors, and improved overall consistency across figures.
> >
> > We kindly ask the reviewers to take this into consideration in reviewing their scores in light of the substantial improvements made. We are grateful for the thoughtful and constructive feedback, which significantly strengthened the clarity and quality of our paper.

---

### Official Review · Reviewer_nen4 · 2025-10-30

**Soundness:** 2
**Presentation:** 2
**Contribution:** 2
**Rating:** 4
**Confidence:** 3

**Summary:**

The paper proposes SPECTRA, a spectral, target-aware augmentation pipeline for imbalanced molecular property regression

**Strengths:**

1. The proposed method is well motivated and clear.
2. The method seems to indeed expend chemical diversity and improving generalization in low-density target regions.
3. The authors provides convincing ablations studies for the method.

**Weaknesses:**

1. The method is very much domain chemistry-specific and was not shown to be generally useful outside of the specific datasets and settings discussed in this work, which makes me doubt whether ICLR is the best venue for this work.
2. There is no running time analysis, and I believe the FGW alignment and spectral decompositions makes the method very heavy computationally. It would be good if the authors would provide some analysis on that and comparison to other methods
3. The empirical part is limited as only only small MoleculeNet regression tasks are used.

**Questions:**

1. Can you provide running time comarision?

---

> ### Author Response · Authors · 2025-11-21
> **Response to reviwer**
>
> Thank you for all the amazing questions and feedback that helped improve our work.
>
> Weakness:
>
> 1. Thank you for the thoughtful feedback. Concerning the chemistry-specific focus of our paper, we would like to underscore that this field has, due to its particular constraints and challenges, been the motivation to significant advances in machine learning and representation learning methods. For that reason, ICLR has been a reference venue for applications in chemistry and drug discovery, as reflected in several papers from ICLR 2025, which aligns well with the interdisciplinary nature of our contribution. As such, we would argue that ICLR is in fact an excellent venue to continue exploring the intersection of machine and representation learning, and chemistry. (Contribution)
>
> 2. We also appreciate the comment regarding the clarity of the runtime analysis. We acknowledge that the original MAE × Time figure was difficult to interpret, and we have replaced it with a clearer boxplot comparison (Figure 6). This updated visualization more effectively shows that our method achieves competitive runtime and offers a favorable balance between predictive accuracy and computational efficiency. In general, it achieves comparable performance with SGIR, and it is 2-5X faster, with a runtime comparable with more time-efficient methods (Presentation/Soundness).
>
> 3. To strengthen the empirical evaluation, we are also running our model on the larger QM9 dataset, which is a molecular regression benchmark directly aligned with the scope of our work. These additional results will help further demonstrate the robustness and scalability of the proposed approach. (Soundness)
>
> Questions
>
> 1. We have modified the original image with runtime analysis (Figure 6). This updated visualization shows that our method achieves competitive runtime and offers a favorable balance between predictive accuracy and computational efficiency. In general, it achieves comparable performance with SGIR, and it is 2-5X faster, with a runtime comparable with more time-efficient methods (Presentation/Soundness).

---

> > ### Author Response · Authors · 2025-11-26
> > **New results**
> >
> > **Contribution**. We have finalized new experiments on the large QM9 dataset (133,885 molecules), further reinforcing the soundness and scalability of our approach:
> > * In terms of predictive performance, our method remains significantly better in predictive performance w.r.t. existing baselines except SGIR.
> > * In relation to SGIR, we still obtain comparable performance, but our approach offers a substantial efficiency advantage, achieving an average 4X faster runtime and providing a better balance between performance and computational efficiency.
> > * Also, on the significance of our method generating chemically plausible and potentially applicable in real-world discovery settings – another distinctive factor w.r.t. SGIR – the generated samples by SPECTRA in this new dataset still fall within Lipinski-compliant ranges, exhibit high uniqueness (0.928) and novelty (0.819), while maintaining structural complexity distributions that closely match those of the original molecules.

---

> > ### Author Response · Authors · 2025-11-26
> > **Presentation Improvements**
> >
> > **Presentation**: We acknowledge the presentation issues raised by the reviewers and have revised the paper accordingly.
> > * We updated the main figure (Figure 2) by removing equations and redesigning it as a cleaner, more diagrammatic illustration that clearly highlights each step of our approach.
> > * Figure 1 was also improved by adding a dataset example and presenting the distributions more clearly.
> > * We consolidated the original tables into a single, clearer table, and in Figure 3, we now color the samples by their target values to better illustrate how the augmented space compares to the original distribution.
> > * We also expanded and refined the ablation table to show performance differences across a broader set of methodological combinations, including augmentation, interpolation, alignment, and the KDE-based inverse distribution.
> > * Finally, we corrected missing symbol explanations in the methodology section, fixed minor errors, and improved overall consistency across figures.
> >
> > We kindly ask the reviewers to take this into consideration in reviewing their scores in light of the substantial improvements made. We are grateful for the thoughtful and constructive feedback, which significantly strengthened the clarity and quality of our paper.

---

### Official Review · Reviewer_Uh3y · 2025-10-31

**Soundness:** 3
**Presentation:** 3
**Contribution:** 3
**Rating:** 6
**Confidence:** 4

**Summary:**

This paper primarily addresses the data imbalance problem in molecular property prediction. It proposes a novel framework called SPECTRA. The core idea of ​​SPECTRA is to generate new, realistic molecular maps in the spectral domain of the graph, specifically designed to "fill" sparse data regions. In this way, it aims to improve the model's prediction accuracy on imbalanced data, particularly on rare target values, without sacrificing overall prediction accuracy.

**Strengths:**

1. Using the concept of geometry-aware graph matching, a geometry-aware and feature-aware correspondence is established between nodes of two different molecular graphs.

2. An innovative spectral domain interpolation method is used to interpolate the eigenvalues ​​and eigenvectors of the Laplacian matrix, as well as node features, ensuring sufficient validity of the generated molecules.

3. Kernel density estimation is used to analyze the distribution of target labels in the training set to identify which regions are "sparse," thereby generating new samples concentrated in areas with the least data.

**Weaknesses:**

1. The paper doesn't seem to explicitly compare its performance with the mentioned "embedding-based enhancement methods."

2. Limited performance. The proposed method doesn't show a significant advantage over the baselines presented, especially for SGIR.

3. Scalability is questionable. To my knowledge, the FGW alignment and spectral decomposition used in the paper both have O(n^3) complexity. The three datasets used in the paper are all very small, raising doubts about whether the method can be scaled to large datasets with millions of compounds (such as ZINC).

**Questions:**

1. The article uses linear interpolation to generate labels, which is a strong assumption. Is it possible to use a trained model, similar to SGIR, to generate pseudo-labels for new molecules to capture the non-linear relationship between molecular structure and labels?

2. The article mentions 100% effectiveness, but also states that graphs that cannot be cleaned by RDKIT will be marked as invalid. Does this effectiveness refer to the effectiveness of all molecules being cleaned by RDKIT, or only the effectiveness of the remaining molecules after discarding those that cannot be cleaned?

---

> ### Author Response · Authors · 2025-11-21
> **Response to reviwer**
>
> We would like to thank all the great feedback that has been provided. These are all helping in improving the paper.
>
> Weakness:
> 1. We appreciate the reviewer’s comment regarding embedding-based augmentation methods. SGIR is indeed an embedding-space augmentation technique, and our work does not currently include a broad comparison with other embedding-based strategies because, to the best of our knowledge, such methods are still limited for regression tasks.
>
> 2. Importantly, although SGIR achieves competitive performance, our method offers two significant advantages that are not captured by SGIR alone. First, it is 2–5× faster computationally across datasets. Second, unlike embedding-space augmentation, our approach generates valid new molecular structures, which provides direct value for downstream virtual screening and molecular design—a different and complementary contribution.
>
> 3. Regarding scalability, we acknowledge that the spectral decomposition introduces additional complexity (as shown in Table 3, an updated table from Appendix). Nonetheless, the overall runtime remains comparable to baseline models, and we are currently evaluating the method on the larger QM9 dataset (133,885 molecules) to better illustrate its scalability in practice. This is also a valuable observation for future work, and we have already included in the manuscript a discussion on exploring more efficient variants of FGW alignment and spectral decomposition in an extension, further reducing computational cost.
>
> Questions:
>
> 1. We agree that using a trained model to generate pseudo-labels, as done in SGIR, can capture nonlinear structure-property relationships. However, our goal in this work is fundamentally different. We designed spectral augmentation to be model-agnostic: it does not rely on any predictive model at generation time, and therefore does not inherit or amplify the biases of a specific regressor. Using a trained model to assign pseudo-labels would make the augmentation strongly dependent on the teacher model’s errors and uncertainties, limiting its generality. Linear interpolation was chosen intentionally because it preserves the geometry of the spectral manifold without imposing additional assumptions from external predictors. It also allows the augmented samples to be used with any downstream model, avoiding circularity (i.e., generating synthetic data using the same model that will later train on it). For these reasons, we opted not to use pseudo-labeling. However, we see value in combining spectral augmentation with model-based label assignment, we have added this suggestion in our plan to explore hybrid strategies in future work.
>
> 2.  We have clarified this issue in the paper based on your feedback. It refers only to the effectiveness of the remaining molecules after discarding those that cannot be cleaned.

---

> > ### Author Response · Authors · 2025-11-26
> > **New Results**
> >
> > **Contribution**. We have finalized new experiments on the large QM9 dataset (133,885 molecules), further reinforcing the soundness and scalability of our approach:
> > * In terms of predictive performance, our method remains significantly better in predictive performance w.r.t. existing baselines except SGIR.
> > * In relation to SGIR, we still obtain comparable performance, but our approach offers a substantial efficiency advantage, achieving an average 4X faster runtime and providing a better balance between performance and computational efficiency.
> > * Also, on the significance of our method generating chemically plausible and potentially applicable in real-world discovery settings – another distinctive factor w.r.t. SGIR – the generated samples by SPECTRA in this new dataset still fall within Lipinski-compliant ranges, exhibit high uniqueness (0.928) and novelty (0.819), while maintaining structural complexity distributions that closely match those of the original molecules.

---

> > ### Author Response · Authors · 2025-11-26
> > **Presentation Improvements**
> >
> > **Presentation**: We acknowledge the presentation issues raised by the reviewers and have revised the paper accordingly.
> > * We updated the main figure (Figure 2) by removing equations and redesigning it as a cleaner, more diagrammatic illustration that clearly highlights each step of our approach.
> > * Figure 1 was also improved by adding a dataset example and presenting the distributions more clearly.
> > * We consolidated the original tables into a single, clearer table, and in Figure 3, we now color the samples by their target values to better illustrate how the augmented space compares to the original distribution.
> > * We also expanded and refined the ablation table to show performance differences across a broader set of methodological combinations, including augmentation, interpolation, alignment, and the KDE-based inverse distribution.
> > * Finally, we corrected missing symbol explanations in the methodology section, fixed minor errors, and improved overall consistency across figures.
> >
> > We kindly ask the reviewers to take this into consideration in reviewing their scores in light of the substantial improvements made. We are grateful for the thoughtful and constructive feedback, which significantly strengthened the clarity and quality of our paper.

---

### Official Review · Reviewer_RUhG · 2025-11-03

**Soundness:** 3
**Presentation:** 2
**Contribution:** 2
**Rating:** 4
**Confidence:** 4

**Summary:**

This manuscript proposes a spectral target-aware graph augmentation framework for imbalanced molecular property regression named SPECTRA, which tries to address the imbalanced regression problem in graph learning. Enhancing samples in sparsely populated regions of the label space while preserving the complex topology and chemical properties of molecular graphs, overcoming the limitations of existing oversampling techniques in regression. Simultaneously, SPECTRA can generate molecules with authentic chemical properties, enhancing interpretability while significantly reducing computational costs compared to existing techniques.

**Strengths:**

This work innovatively applies spectral graph augmentation to molecular property regression. By preserving the topological structure and chemical properties of molecular graphs while enhancing samples in sparsely populated regions of the label space, it overcomes the limitations of existing oversampling techniques in regression. The methodology is well designed, with a comprehensive workflow that includes spectral decomposition, FGW alignment, eigenvalue interpolation, and graph reconstruction. Its effectiveness has been validated across multiple datasets and has been comprehensively evaluated using a variety of metrics.

**Weaknesses:**

The absence of statistical significance tests and experiments, such as quality analysis of the generated molecules, could be addressed by incorporating relevant experiments and analyses to enhance the persuasiveness of the findings. Regarding writing and presentation, the paper exhibits numerous minor errors in the formatting of figures and tables.

**Questions:**

1. The specific interpolation process depicted in Figure 2 is presented through formulas rather than diagrams, making it less intuitive.
2. In the formulas presented in the Method chapter of this manuscript, the meanings of certain parameters and symbols have not been explicitly explained.
3. In Section 3.2, the formula for the FGW states that the alpha parameter balances structural similarity and feature similarity, but the experiments did not analyze the impact of the alpha parameter on performance.
4. The article's formatting contains numerous errors. For example, in Section 4.1, the text references Figure 3, but the caption beneath the corresponding figure displays Table 3. This error has subsequently led to further issues in the formatting of figure and table captions throughout the text. The authors are urged to adopt a more conscientious approach, thoroughly review the content, and make the necessary corrections.
5. Table 1 shows that all generated molecules are chemically valid (100% validity), but it should be demonstrated that the generated molecules possess practical application potential. The authors are advised to add quality analysis of the generated molecules.
6. In the explanation of Figure 3 in Section 4.2, it is stated that enhanced molecules are represented by crosses, but the figure uses circles identical to those of the original molecules. The authors are requested to carefully revise this section and strive to avoid such basic errors in the manuscript.
7. Figure 4 illustrates the Mean Absolute Error (MAE) across various target intervals; however, statistical test results are not reported. It is advisable to include an analysis of statistical significance.
8. The SGIR model appears to outperform SPECTRA with lower MAE scores across three distinct datasets. Analyze the reasons for this and explore further.
9. How can we analyze from Figure 5 that SPECTRA achieves a trade-off between performance and efficiency? Please further optimize the illustrated results to make them more intuitive and understandable, while ensuring the analysis is more profound and reliable.
10. The Appendix mentions the hyperparameter search range but does not list the final selected values. Could you provide a table of optimal configurations?

---

> ### Author Response · Authors · 2025-11-21
>
> We want to thank you for your detailed reviews, which definitely improved our paper.
>
> **Weakness**
>
> * We agree that statistical tests should be added, and we have done so in the Appendix, where we demonstrate that the performance differences relative to SPECTRA are statistically significant (Soundness).
> * In Section 4.1, we strengthened our analysis of the generated molecules. In addition to our original results, showing that the molecules are unique and novel (Table 1), expand scaffold diversity while remaining within chemically reasonable bounds (Table 1 and the new Figure 7 in the Appendix), and follow similar property distribution trends, we now improve Figure 4 by adding an analysis based on Lipinski’s Rule of Five. This widely used guideline in drug discovery evaluates physicochemical properties relevant to oral bioavailability, and our results show that the generated molecules fall within accepted ranges (Soundness). The SAS and QED, which are also very important, were moved to the Appendix.
> * We also reviewed the manuscript carefully and corrected the minor presentation issues noted by the reviewer (Presentation).
>
> [1] BG Gimenez, MS Santos, M Ferrarini, and JPS Fernandes. Evaluation of blockbuster drugs under the rule-of-five. Die Pharmazie-An International Journal of Pharmaceutical Sciences, 65(2):148–152, 2010.
>
> **Questions**
>
> 1.  We agree, and we have improved the figure to make the process more intuitive. The updated version illustrates how interpolation is performed from the aligned eigenvalues and eigenvectors after GW/FGW-based Laplacian alignment using the graph structure. We also simplified the presentation by removing unnecessary formulas and adopting a more diagram-like layout to enhance clarity  (Presentation).
>
> 2. Thank you so much for your attention to the details of the paper. The explanations of the missing parameters and symbols were added to the paper. (Presentation)
>
> 3. We agree that this analysis was important and missing from the original version. We have now added it to the Appendix (Figure 9), showing the validation performance across different values of alpha and gamma. The results indicate that these parameters strongly influence performance, with the best results typically obtained for 0.25≤alpha≤0.70 and gamma∈{0.1,0.5}. Showing that a better balance in structural information improves performance while aligned with the mixing coefficient.
>
> 4.  Again, we appreciate the detailed observations, which show that it is indeed improving significantly. We already did the correction (Presentation).
>
> 5.  As commented previously, in Section 4.1, we strengthened our analysis of the generated molecules. In addition to our original results, showing that the molecules are unique and novel (Table 1), expand scaffold diversity while remaining within chemically reasonable bounds (Table 1 and the new Figure 7 in the Appendix), and follow similar property distribution trends, we now improve Figure 4 by adding an analysis based on Lipinski’s Rule of Five. This widely used guideline in drug discovery evaluates physicochemical properties relevant to oral bioavailability, and our results show that the generated molecules fall within accepted ranges, demonstrating their practical applications (Soundness). The SAS and QED, which are also very important, were moved to the Appendix.
>
> 6.  Thank you again for the observation, and this was updated along with the update of the image to present the boundaries for Lipinski's Rule of Five (Presentation, Soundness).
>
> 7.  This is indeed a critical analysis, and we thank the reviewer for the suggestions. We have added these tests in the Appendix, where we demonstrate that the performance differences relative to SPECTRA are statistically significant, showing that SPECTRA outperformed most of the methods(Soundness).
>
> 8.  Concerning the perceived advantage of SGIR over SPECTRA, we have added statistical tests (Table 6) showing that the differences between SGIR and SPECTRA are minimal. The slight advantage of SGIR comes from the benefits of pseudo-labeling real unlabeled molecules, while SPECTRA generates new chemically-valid samples purely from learned spectral representations, which, although similar in performance, provides greater real-world significance. (Contribution)
>
> 9.  We agree and have improved Figure 6  for a more comprehensive understanding of runtime. This updated visualization more effectively shows that our method achieves competitive runtime and offers a favorable balance between predictive accuracy and computational efficiency. In general, it achieves comparable performance to SGIR and is 2-5X faster, with a runtime comparable to that of more time-efficient methods (Soundness).
>
> 10. We have added the current optimal hyperparameter values to the paper. In addition, we are running experiments on a larger dataset and will update the hyperparameter settings accordingly in the revised version.

---

> > ### Author Response · Authors · 2025-11-26
> > **New results**
> >
> > **Contribution**. We have finalized new experiments on the large QM9 dataset (133,885 molecules), further reinforcing the soundness and scalability of our approach:
> > * In terms of predictive performance, our method remains significantly better in predictive performance w.r.t. existing baselines except SGIR.
> > * In relation to SGIR, we still obtain comparable performance, but our approach offers a substantial efficiency advantage, achieving an average 4X faster runtime and providing a better balance between performance and computational efficiency.
> > * Also, on the significance of our method generating chemically plausible and potentially applicable in real-world discovery settings – another distinctive factor w.r.t. SGIR – the generated samples by SPECTRA in this new dataset still fall within Lipinski-compliant ranges, exhibit high uniqueness (0.928) and novelty (0.819), while maintaining structural complexity distributions that closely match those of the original molecules.

---

> > ### Author Response · Authors · 2025-11-26
> > **Presentation Improvements**
> >
> > **Presentation**: We acknowledge the presentation issues raised by the reviewers and have revised the paper accordingly.
> > * We updated the main figure (Figure 2) by removing equations and redesigning it as a cleaner, more diagrammatic illustration that clearly highlights each step of our approach.
> > * Figure 1 was also improved by adding a dataset example and presenting the distributions more clearly.
> > * We consolidated the original tables into a single, clearer table, and in Figure 3, we now color the samples by their target values to better illustrate how the augmented space compares to the original distribution.
> > * We also expanded and refined the ablation table to show performance differences across a broader set of methodological combinations, including augmentation, interpolation, alignment, and the KDE-based inverse distribution.
> > * Finally, we corrected missing symbol explanations in the methodology section, fixed minor errors, and improved overall consistency across figures.
> >
> > We kindly ask the reviewers to take this into consideration in reviewing their scores in light of the substantial improvements made. We are grateful for the thoughtful and constructive feedback, which significantly strengthened the clarity and quality of our paper.

---

### Author Response · Authors · 2025-11-21
**General Improvements**

We want to thank all reviewers for their comments and insights. They were instrumental in identifying areas that needed improvement or clarification, and we believe the paper is significantly stronger because of your contributions. Because your reviews led to overall improvements, we thought it would be a good idea to compile these changes for you all.

**Soundness:** We significantly strengthened the empirical and methodological rigor of the paper:
* Added statistical significance tests comparing all methods against SPECTRA (Appendix, Table 6), demonstrating that SPECTRA’s improvements are statistically significant across datasets.
* Added hyperparameter sensitivity analysis for alpha (balance between feature and structural information) and gamma (mixing parameter) (Appendix, Figure 9).. The results indicate that these parameters strongly influence performance, with the best results typically obtained for 0.25≤alpha≤0.70 and gamma∈{0.1,0.5}. Showing that a better balance in structural information improves performance while aligned with mixing coefficient.
* In Section 4.1, we strengthened our analysis of the generated molecules. In addition to our original results-showing that the molecules are unique and novel (Table 1), expand scaffold diversity while remaining within chemically reasonable bounds (Table 2 and the new Figure 7 in the Appendix), and follow similar property distribution trends-we now improve Figure 4 by adding an analysis based on Lipinski’s Rule of Five. This widely used guideline in drug discovery evaluates physicochemical properties relevant to oral bioavailability, and our results show that the generated molecules fall within accepted ranges (Soundness).
* We are running new experiments for a large dataset QM9 (133,885 molecules).
* We have also moved the ablation study from Appendix to the main section, demonstrating the impact of each step of our model architecture.

**Presentation:** We improved clarity, readability, and correctness throughout the manuscript:
* Updated Figure 2 with a clearer diagram-like illustration of the spectral interpolation process, removing unnecessary formulas.
* Improved Figure 6 for a more comprehensive understanding of runtime. This updated visualization more effectively shows that our method achieves competitive runtime and offers a favorable balance between predictive accuracy and computational efficiency. In general, it achieves comparable performance with SGIR, and it is 2-5X faster, with a runtime comparable with more time-efficient methods.
* We also clarified the description of network alignment in the methods section. In its basic form, network alignment (or graph matching) seeks a correspondence between nodes in two graphs that best preserves their structural relationships [1]. In chemistry, this is commonly used to assess structural similarity across molecules [2].Without graph alignment, the node indices of the two graphs do not correspond, causing their eigenvectors to represent unrelated structural regions. Interpolating unaligned spectra produces Laplacians that no longer reflect meaningful graph structure, leading to invalid or chemically inconsistent molecules after reconstruction. Alignment is therefore essential to ensure that interpolation operates on comparable structural components.

**Contribution:** We clarified the novelty and broader value of the work.
* Articulated that SPECTRA is a model-agnostic structural augmentation method that generates valid new molecules, adding value for virtual screening and molecular design—distinct from embedding-based approaches like SGIR that augment without generating valid structures (molecules).
* In addition to this, highlighting that SPECTRA achieves competitive performance with SGIR while offering significantly better runtime (2-5X faster).

[1].Lázaro, T., Guimerà, R. & Sales-Pardo, M. Probabilistic alignment of multiple networks. Nat Commun 16, 3949 (2025). https://doi.org/10.1038/s41467-025-59077-7

[2].Emmert-Streib, F., Dehmer, M. & Shi, Y. Fifty years of graph matching, network alignment and network comparison. Inf. Sci. 346-347, 180–197 (2016).

---

> ### Author Response · Authors · 2025-11-26
> **New Results**
>
> **Contribution**. We have finalized new experiments on the large QM9 dataset (133,885 molecules), further reinforcing the soundness and scalability of our approach:
> * In terms of predictive performance, our method remains significantly better in predictive performance w.r.t. existing baselines except SGIR.
> * In relation to SGIR, we still obtain comparable performance, but our approach offers a substantial efficiency advantage, achieving an average 4X faster runtime and providing a better balance between performance and computational efficiency.
> * Also, on the significance of our method generating chemically plausible and potentially applicable in real-world discovery settings – another distinctive factor w.r.t. SGIR – the generated samples by SPECTRA in this new dataset still fall within Lipinski-compliant ranges, exhibit high uniqueness (0.928) and novelty (0.819), while maintaining structural complexity distributions that closely match those of the original molecules.

---

> ### Author Response · Authors · 2025-11-26
> **Presentation Improvements**
>
> **Presentation**: We acknowledge the presentation issues raised by the reviewers and have revised the paper accordingly.
> * We updated the main figure (Figure 2) by removing equations and redesigning it as a cleaner, more diagrammatic illustration that clearly highlights each step of our approach.
> * Figure 1 was also improved by adding a dataset example and presenting the distributions more clearly.
> * We consolidated the original tables into a single, clearer table, and in Figure 3, we now color the samples by their target values to better illustrate how the augmented space compares to the original distribution.
> * We also expanded and refined the ablation table to show performance differences across a broader set of methodological combinations, including augmentation, interpolation, alignment, and the KDE-based inverse distribution.
> * Finally, we corrected missing symbol explanations in the methodology section, fixed minor errors, and improved overall consistency across figures.
>
> We kindly ask the reviewers to take this into consideration in reviewing their scores in light of the substantial improvements made. We are grateful for the thoughtful and constructive feedback, which significantly strengthened the clarity and quality of our paper.

---

### Note · Program_Chairs · 2026-01-17
**Submission Desk Rejected by Program Chairs**

The following references in this submission do not refer to real documents and/or have major errors in bibliographic information:

 Zhaoping Xiong, Dingyan Wang, Xiaohong Liu, Feisheng Zhong, Xutong Wan, Xiang Li, Zhaojian Li, Xiaomin Luo, Kaixian Chen, Hualiang Jiang, et al. Attentive fp: Augmenting graph neural networks with attentive message passing for molecular property prediction. Journal of Chemical Information and Modeling, 60(6):2213-2228, 2020.